



# Structure of the Central Sumatran Subduction Zone Revealed by Local Earthquake Travel Time Tomography Using Amphibious Data

5   Dietrich Lange[1], Frederik Tilmann[2,6], Tim Henstock[3],
Andreas Rietbrock[4], Danny Natawidjaja[5] and Heidrun Kopp[1,7]

[1] GEOMAR Helmholtz Centre for Ocean Research Kiel, Germany.
[2] Helmholtz-Zentrum Potsdam, Deutsches GeoForschungsZentrum GFZ, Germany.
10  [3] University of Southampton, United Kingdom.
[4] University of Liverpool, Liverpool, UK.
[5] LabEarth, Indonesian Institute of Sciences (LIPI).
[6] Freie Universität Berlin, Berlin, Germany.
[7] Christian-Albrechts-Universität zu Kiel, Germany.

*Correspondence to*: Dietrich Lange (email: dlange@geomar.de)

**Abstract.**

The Sumatran subduction zone exhibits strong seismic and tsunamogenic potential with the prominent examples of the 2004,
2005 and 2007 earthquakes. Here, we invert travel time data of local earthquakes for $v_p$ and $v_p/v_s$ velocity models of the
20  central Sumatran forearc. Data were acquired by an amphibious seismometer network consisting of 52 land stations and 10
ocean bottom seismometers located on a segment of the Sumatran subduction zone that had not ruptured in a great
earthquake since 1797 but witnessed recent ruptures to the north in 2005 (Nias earthquake, Mw=8.7) and to the south in
2007 (Bengkulu earthquake, Mw=8.5). 2D and 3D $v_p$ velocity anomalies reveal the downgoing slab and the sedimentary
basins. Although the seismicity pattern in the study area appears to be strongly influenced by the obliquely subducting
25  Investigator Fracture Zone to at least 200 km depth, the 3D velocity model shows prevailing trench parallel structures at
depths of the plate interface. The tomographic model suggests a thinned crust below the basin east of the forearc islands
(Nias, Pulau Batu, Siberut) at ~180 km distance to the trench. $V_p$ velocities beneath the magmatic arc and the Sumatran fault
zone SFZ are around 5 km/s at 10 km depth and the $v_p/v_s$ ratios in the uppermost 10 km are low, indicating the presence of
felsic lithologies typical for continental crust. We find moderately elevated $v_p/v_s$ values of 1.85 at ~150 km distance to the
30  trench in the region of the Mentawai fault. $V_p/v_s$ ratios suggest absence of large scale alteration of the mantle wedge and
might explain why the seismogenic plate interface (observed as a locked zone from geodetic data) extends below the
continental forearc Moho in Sumatra. Reduced $v_p$ velocities beneath the forearc basin covering the region between Mentawai
Islands and the Sumatra mainland possibly reflect a reduced thickness of the overriding crust.





## 1 Introduction

The largest earthquakes on Earth occur along subduction zones where the oceanic plate is subducted beneath an upper continental plate and large stress is accumulated during the interseismic phase of the seismic cycle. Offshore Sumatra, the oceanic Indo-Australian plate subducts obliquely beneath the Eurasian plate (Fig. 1). In the last decade, the margin hosted a

number of great earthquakes on the subduction thrust (Aceh-Andaman 26 December 2004 Mw=9.2, Nias 28 March 2005 Mw=8.6, Bengkulu 12 September 2007 Mw=8.5). Additionally, major events such as the intermediate depth Mw=7.6 earthquake of 30 September 2009 (e.g. McCloskey et al. 2010, Wiseman et al. 2012) and the shallow and slow rupture of the 25 October 2010 Mentawai tsunami earthquake (Mw=7.8) (Bilek et al. 2011, Lay et al. 2011, Newman et al. 2011) were associated with significant loss of life. Yet, a part of the margin near the northern Mentawai Islands (below Siberut) remains

unbroken since 1797 (Newcomb & McCann 1987, Natawidjaja et al. 2006, Konica et al. 2008, Chlieh et al. 2008, McCloskey et al. 2010). The region is strongly coupled as inferred from GPS observations and coral data (Chlieh et al. 2008). Further to the south, geodetic records suggest that only half of the interseismic tectonic strain accumulated since the great earthquake of 1833 (Fig. 1) might have been released by the 2007 Bengkulu earthquake (Konca et al. 2008). Sieh et al. (2008) estimate the slip deficit below Siberut Island since the large ruptures of 1797 and 1833 to be ~8 m and a reduced slip

deficit of ~5 m for the Batu Islands due to the lower degree of coupling. Therefore, the segment is in an advanced stage of the seismic cycle, although east of Siberut Island there has been significant intra-slab seismic activity, including the Mw=7.6 Padang earthquake on 30 September 2009 (Fig. 2) at intermediate depth (~85 km), which caused significant damage in the city of Padang. Based on Coulomb stress modelling, McCloskey et al. (2010) suggest that the 2009 Padang earthquake did not significantly relax the accumulated stress on the Mentawai segment leaving the threat of a great tsunamogenic

earthquake on the Mentawai patch below Siberut Island is undebated (e.g. Konca et al. 2008, Sieh et al. 2008).

The down-dip limit of subduction thrust earthquakes was suggested to be a function of temperature at the plate interface and to be controlled by the transition from unstable to stable sliding along the plate interface (e.g. Tichelaar and Ruff, 1993). Hyndman et al. (1997) estimate the maximum temperature for seismic behaviour to be 350°C, while large earthquakes may

propagate with decreasing slip down to the 450°C isotherm. An additional limiting factor of the seismogenic zone results from the presence of hydrated minerals (serpentinite) in the forearc mantle wedge, suggesting that the down dip limit of the seismogenic zone correlates to the upper plate Moho (Oleskevich et al., 1999). However, for the Sumatran margin the seismogenic zone is suggested to reach below the continental Moho based on gravity surveys and wide-angle refraction and local earthquake tomography (Siberut: Simoes et al. 2004; Aceh basin and Simeulue: Dessa et al. 2009; Klingelhoefer et al.

2010, Tilmann et al. 2010; Southern Mentawai Islands: Collings et al. 2012). For central Sumatra Chlieh et al. (2008) estimate the width of the seismogenic zone based on geodetic data between 20 km and 50 km, with the largest width approximately alongside Siberut, and the smallest widths at the Batu Islands and between Sipora and the Pagai Islands.




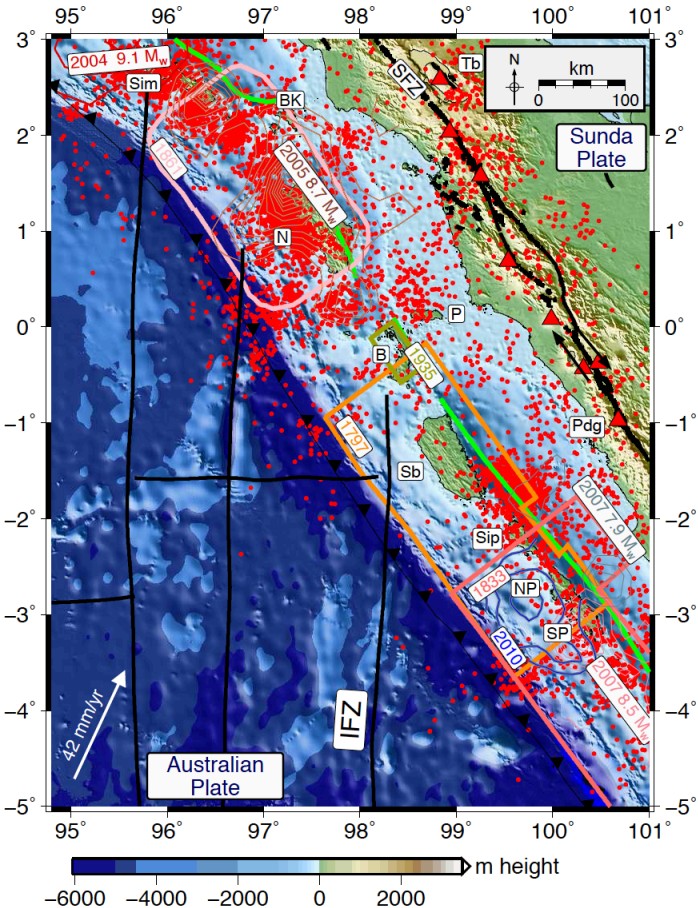

**Figure 1.** Map showing the tectonic setting of the central Sumatran subduction zone. The locations of earthquakes are indicated by red circles (NEIC catalogue, M≥6, 01/01/1990 until 01/09/2017). The Mentawai Fault (Diament et al. 1992) (green line offshore) and the Sumatran Fault (Sieh and Natawidjaja 2000) (black line onshore) are also shown. Bathymetry and topography from the SRTM plus database (Becker et al. 2009). Oceanic fracture zones from Cande et al. (1989) and Tang et al. (2013). Rupture zones of the great 1797 and 1833 earthquakes are based on uplift of coral micro-atolls (Natawidjaja et al. 2006). Rupture areas of the 1861, 1935 and 1984 earthquakes are given by Rivera et al. (2002). Slip distribution of 2004 earthquake from Chlieh et al. (2007). Slip distribution of the 2005 and 2007 earthquakes from Konca et al. (2007, 2008). Convergence between the Australian Plate and the forearc sliver from McNeill and Henstock (2014). Volcanoes (Smithsonian Institute) shown with red triangles. Abbrevations: Sim: Simeulue; BK: Banyak Islands; Tb: Toba; N: Nias; B: Batu Islands; P: Pulau Pini; Sb: Siberut Island; Sip: Sipora; NP: North Pagai; SP: South Pagai; Pdg: Padang.

Previous local earthquake tomography studies were conducted in northern Sumatra focussing on the crustal structure of the region around Lake Toba (Masturyono et al., 2001, Koulakov et al., 2009, Stankiewicz et al., 2010, Koulakov et al., 2016) or on the shallow crustal structure along the Sumatran Fault (Muksin et al. 2013). Pesicek et al. (2010) imaged the deeper slab geometry including the upper-mantle and transition along the Sumatra, Andaman and Burma subduction zones using a regional-global body wave tomography. Offshore, the tomography study of Collings et al. (2012) resolves the deeper

structure beneath north and south Pagai where the 25 October 2010 tsunamogenic event occurred. Structural information is known from active seismic reflection and refraction studies for a significant number of profiles (e.g. Franke et al., 2008; Dean et al, 2010; Klingelhoefer et al. 2010, Mukti et al., 2012, Shulgin et al 2013). The Mentawai fault (Diament et al.,





1992), located between the forearc islands and the mainland, was recently imaged as a southwest dipping backthrust (e.g. Singh et al., 2010, Wiseman et al., 2011). However, there is only limited information about the deep forearc structure and the seismogenic zone (down to depths of ~50 km) of the central Sumatran margin.

Offshore central Sumatra a ~2500 km long NS trending topographic feature, the Investigator Fracture zone (IFZ), is situated on the incoming Indo-Australian plate and is subducted at a rate of 57 mm/yr below the Sumatran mainland (Fig. 1). Seismicity occurring in the prolongation of the IFZ down to depths of 200 km presumably reflects the subducting trace of the IFZ (Fauzi et al., 1996, Lange et al., 2010). At shallower depths beneath the Batu Islands both the forearc crust and the plate interface are characterised by enhanced seismicity levels with a number of persistent clusters. This region hosted several

major events during the last century (e.g. 1935 Mw=7.7 and 1984 Mw=7.2; Rivera et al., 2002) but was not affected by great earthquakes in the last 220 years at least (Konca et al., 2008). Together with the decreased locking this justifies its identification as a persistent segment barrier (Natawidjaja et al. 2006).

The development of the forearc basin between the Sumatran mainland and the Island of Nias was described in Matson &

Moore (1992). Overall, the Sumatran margin is characterized by rapid accretion since the early Oligocene with current trench fill ages from Quaternary to Eocene ages (Moore et al., 1982). The uplift rates of Nias slope sediments is suggested to be in the order of 100-300 meters/my and accreted material has been uplifted by more than 800 m in the centre of Nias Island (Moore et al., 1980).

In order to investigate the deep structure of the central Sumatran subduction zone a dense, temporary, amphibious (on-offshore) seismic network was installed in central Sumatra in 2008. Besides local seismicity, the main targets of the seismometer network was to obtain velocity models of the complete marine and continental forearc in order to decipher downdip and along-strike structural variations of the Sumatran subduction zone.

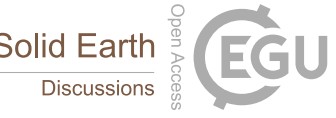

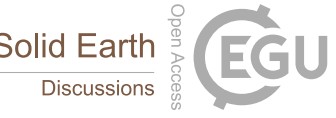

**Figure 2.** Station and event distribution and ray coverage of the inversion. Green circles indicate events used in the inversion and corresponding raypaths are indicated by grey lines. Blue circles indicate the complete local catalogue (e.g. with events outside the network, or excluded for other reasons such as large RMS or a small number of picks per event). Yellow triangles (grey triangles for two-week OBS deployment) indicate stations used in the study. The gCMT focal mechanism of the 30 September 2009 Padang earthquake and its aftershocks (McCloskey, 2010) are indicated in red. Light blue squares show the events from the seismic crisis during 2008 occurring in a persistent seismicity cluster SE of Siberut Island (Wiseman et al. 2011), including the Mw 7.2 mainshock of February 25 (blue star). Other symbols as in Fig. 1.

## 2 Earthquake Data

For the local earthquake tomography we use data from a dense amphibious network of up to 62 stations covering the Sumatran Forearc from the trench to the volcanic arc (Lange et al., 2010). 52 land stations from SEIS-UK were installed in April 2008 between 1.8°S and 1.8°N on the mainland and on the islands of Nias, Pulau Batu, Siberut, and North Pagai. Offshore, the network was complemented by ten three-component ocean bottom seismometers (OBS; Minshull et al., 2004) equipped with differential pressure gauges from June 2008 to February 2009. During October 2008, ten land stations were





removed from the Sumatran mainland, leaving the remaining 42 land stations until February 2009. The land stations continuously recorded three spatial components with sample rates of 50 and 100 Hz. We also include data from eight permanent stations operated by BMKG (Meteorological and Geophysical Agency of Indonesia, www.bmkg.go.id), Geofon (http://geofon.gfz-potsdam.de/, FDSN network code 1G) and station GSI and BKNI operated by the GEOFON network

(FDSN network code GE, GEOFON Data Centre (1993)) in the analysis. Furthermore, we include five stations for strong events from a temporary deployment north of our project area (Stankiewicz et al. 2010, GEOFON network code 7A-2008, Ryberg and Haberland, 2008) and three stations from an adjacent temporary network to the south (Collings et al., 2012). Additionally, data from 46 ocean bottom stations (OBS/H) from an active-source experiment offshore (25 May and 10 June 2008) were considered (Vermeesch et al., 2009). A summary of the stations can be found in the supplementary material of

Lange et al. (2010), Table 1. The main sources of noise on the records were tree movement and rain due to the tropical environment, affecting in particular the horizontal components. On the ocean bottom stations, S waves were very difficult to pick because in addition to high noise levels, the onset of S arrivals was usually poorly defined due to basement conversions.

From the original dataset (Lange et al., 2010) with 1,271 events and 32,4781 manually picked arrival times (20,251 P and

12,220 S-onsets), we selected events with more than nine P and four S phase picks and RMS values smaller than 1.5 s. Then, we removed all phase arrivals with residuals larger than 2 s. Because of the large number of stations and events on or near the Sumatran fault zone (SFZ) we applied a more stricter selection criteria for these crustal events (depths less than ~20 km and distances of less than 35 km from the fault trace) by excluding events with less than eleven recording stations and RMS values greater than 1 s. These selection criteria were chosen to improve the numerical balance of events from different parts

of the study region (slab events: 9,165 onsets, SFZ events: 7,686 onsets). Finally, we ignored stations with less than 15 high-quality observations. OBS/H stations with high station residuals or dubious time corrections were not included in the inversion in order to be sure that all the observed travel times are accurate. After having checked the stability of the 2D inversions exclusively with events within the network (largest azimuth range with no observations, GAP<=180°), events with GAP<200° were included in the inversion. We carefully checked that the relaxation of the GAP criterion to 200° did

not produce substantially different velocity models. Figure 2 shows the ray coverage with many paths criss-crossing in the central part of the model. The final dataset consists of 655 events with 9,939 P (therefrom 2,626 with the highest quality, using the quality assignment of Lange et al. (2010)) and 4,859 S-arrivals (626 with highest quality).

### 3 Local earthquake tomography

We invert for 2-D and 3-D velocity models of the Sumatran subduction zone using local earthquake tomography (LET)

techniques (Aki and Lee, 1976; Kissling, 1988) by applying the well-established inversion code SIMUL2000 (Thurber, 1983; Evans et al., 1994) for the simultaneous inversion of hypocentral parameters and velocity structure ($v_p$, $v_p/v_s$). The original algorithm by Thurber (1983) was subsequently modified and enhanced with new features (e.g. Eberhart-Phillips,



1986, 1993; Um and Thurber, 1987; Thurber and Eberhart-Phillips, 1999) and has been widely used for various LET studies (e.g. Graeber and Asch, 1999; DeShon and Schwartz, 2004; Haberland et al., 2009). For the inversion of the Sumatra data (located on the southern and northern hemisphere) SIMUL2000 needed to be modified to operate across the equator.

In the damped least squares inversion, the velocity structure $v_p$ and $v_p/v_s$ are inverted from the observed travel times. The velocity model is represented by velocity values specified on a rectangular grid of irregularly spaced nodes. The velocity for a given point within the grid is calculated by linearly interpolating the eight neighbouring grid nodes. For a fast calculation of the path integral, Thurber (1983) implemented the ray tracer based on the "Approximate Ray Tracing" technique (ART). Receiver and source are connected with different circular arcs with varying radii and inclinations. Then, the 2-D circular arcs
are perturbed in three dimensions to further minimize the travel time in an iterative process (Um and Thurber, 1987). Following common practice we applied a staggered inversion scheme starting with inversions for a one-dimensional model, followed by an inversion for a two-dimensional velocity model, and finally a 3D-inversion using the 2D model as a starting model. For each inversion the arrival times were weighted by their assigned pick uncertainties.

The importance of careful selection of the minimum 1-D model was described by various authors (e.g. Kissling, 1988; Eberhart-Phillips, 1990; Kissling et al., 1994). As 1D $v_p$ starting model we used the "minimum one-dimensional model" from Lange et al. (2010) (Fig. 3, green line), which was obtained from a brute force search of different one-dimensional input models using the program VELEST (Kissling et al., 1994) and an active source studies (Vermeesch et al. 2009). For the inversion of the 2D velocity model we tested different $v_p$ starting models from an active source refraction study
(Vermeesch et al., 2009), from the seismicity study of Lange et al. (2010), and the LET of Collings et al. (2012) (Fig. 3). Based on these different velocity models the inversion of the 2D $v_p$ velocity model leads to very similar results. For the inversion of the 2D $v_p/v_s$ model we fixed (i.e. highly damped) the $v_p$ model and used a constant $v_p/v_s$ ratio of 1.77 derived from Wadati diagrams as starting model.

Horizontal distances between nodes were 30 km in the trench perpendicular direction (x-axis) and, for the 3D inversion, 50 km in the trench-parallel direction (y-axis). In the vertical direction (z-axis) node spacing is 10 km down to 50 km depth with one additional node at 5 km depth. Below 50 km depth coarser node spacing is used with nodes at 70, 90, and 120 km depth to account for the decreasing ray coverage with depth. The grid is rotated relative to the trend of the north direction by 28° and centred at 0°N, 99°E. After testing different spacing parameters for 2D and 3D inversions in all three directions, we
selected the node spacing as a compromise between resolution and stability of the inversion.

Following Evans et al. (1994), one additional node is introduced at all edges of the model with a much larger distance for computational reasons. The damping value of the damped least squares inversion was carefully determined by "trade-off" curves between model variance and data variance (Eberhart-Phillips, 1986) and is chosen such as to simultaneously





minimize the model variance and data variance. This is achieved by plotting model variance versus data variance of one-step inversions with different damping values for a given model geometry. The final 3D inversion yields a significant reduction of the data variance. The P wave data variance reduction is 76% compared to the minimum 1D dimensional model. The S wave data variance reduction is only 18% compared to a homogeneous model with $v_p/v_s$ values of 1.77. The small degree of

5   improvement for the 3D velocity model relates mostly to the high noise levels on the horizontal components resulting in S onsets of low quality.

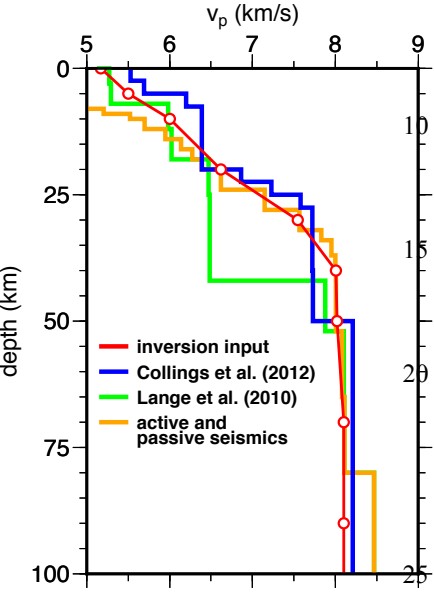

**Figure 3.** Velocity models used as starting model for the two-dimensional tomographic inversion of $v_p$ (red line and red circles, respectively). Because of the large number of stations and events on or near the SFZ the minimum 1D velocity model (green line) is dominated by the crustal structure of the Sumatran mainland. This velocity model is not appropriate for the events in the outer forearc, so we constructed a layered 1D $v_p$ velocity function based on an active source refraction study (Vermeesch et al., 2009). Above 30 km depth, we adopted a one-dimensional, staircase-like velocity depth function based on an active source refraction study (Vermeesch et al., 2009, orange line); for depths greater than 30 km we adopted the one-dimensional velocity function from a minimum 1D velocity model (Lange et al., 2010). The blue line shows the minimum 1D $v_p$ velocity model from Collings et al. (2012) for the region adjacent to the southeast of our study region but covering a similar part of the subduction system.

**4 Resolution and Uniqueness**

**4.1. Dependency of 2D inversion on 1D input model**

30   We tested the dependency of the 2D inversion on the 1-D input model in order to estimate the stability of the inversion and its ability to converge. This was done by constructing (realistic) randomized $v_p$ velocity models with increasing velocity for increasing depths. These models were used as alternative starting models and the inversion was otherwise carried out identically. We also tested alternative $v_p$ starting models from the active source refraction study of Vermeesch et al. (2009) and the minimum 1D $v_p$ model of Collings et al. (2012) (Fig. 3). We then carefully checked the dependency of the 2D

35   inversion on the velocity models and only found minor dependency of the input model, indicating a very stable result of the 2D inversion. For the final inversion we used the velocity model indicated by the red line in Fig. 3.



### 4.2. Spread value

The spread function of the resolution matrix poses a possibility to assess the resolution of the model nodes. The spread function (e.g. Toomey and Foulger, 1989) summarises the information contained in a single averaging vector or row of the full-resolution matrix. For a peaked resolution, i.e. low smearing, the diagonal element is much larger than the off-diagonal

5   elements and the spread is low. The spread values (Fig. 4) show low values in the central part of the model between the SFZ and the islands with a reduced resolution in the region offshore Siberut and Nias. At depths larger than 50 km, resolution is moderate as indicated by reduced spread values to 80 km depth. Below the Wadati-Benioff zone there is basically no penetration and thus no resolution at all.

### 4.3. Checkerboard tests and synthetic recovery tests

10  Synthetic tests and checkerboard tests were carried out to evaluate the resolution of the inversion. The procedure includes forward calculation of the travel times for a synthetic velocity model and the actual source- and receiver distribution. In a second step the calculated travel times are then perturbed with Gaussian noise, with a standard deviation dependent on the pick quality, 0.05 s for the highest-quality observations, to 0.2 s for the lowest-quality observations. Finally, the perturbed travel times are introduced into the inversion.

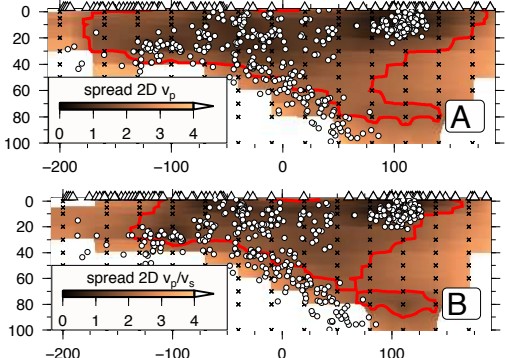

**Figure 4.** Spread values of 2D $v_p$ (panel A) and $v_p/v_s$ (panel B) inversion. Regions with low spread values and thus good resolution (selection of a cut-off spread value based on analysis of the model resolution matrix and synthetic tests) are

20  encircled with a red line. Circles indicate hypocentres and grid nodes are shown with crosses. Stations are indicated with triangles.





### 4.3.1 2D checkerboard tests

2D checkerboard tests were conducted for $v_p$ and $v_p/v_s$ models (Fig. 5), respectively. We used varying block sizes in which the input models were perturbed by ±5% (Fig. 6). At the highest resolution (blocks with one grid point dimension, equivalent to 30 km horizontal space and 10 km vertical space in the shallow part of the model), the pattern of perturbations is restored

5  in the central part but the maximum amplitude of the recovered anomalies was 3.7%, i.e. only about 75% of the input anomalies. The checkerboard tests with 2×2 blocks (60x20 km) and lower resolution restore both the pattern and the amplitudes in the central part of the model and beneath the SFZ.

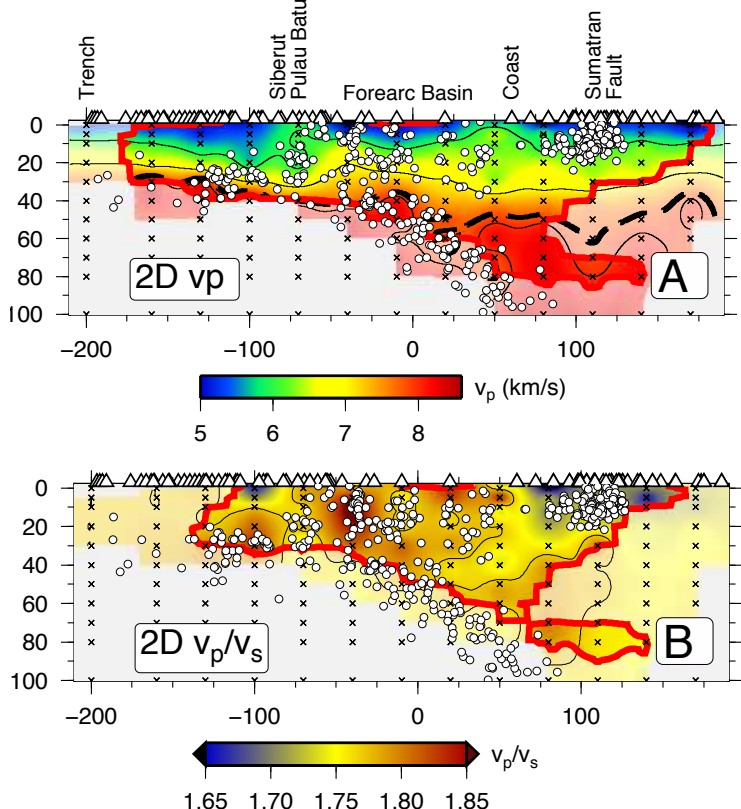

10  **Figure 5.** 2-D tomographic velocity models for $v_p$ and $v_p/v_s$ models (profile direction is trench perpendicular). Regions with good resolution are encircled with a red line. Circles indicate hypocentres and grid nodes are shown with crosses. Stations are indicated with triangles. The dashed line in panel A indicates the $v_p$ 7.8 km/s contour line and is used as a proxy for the Moho.





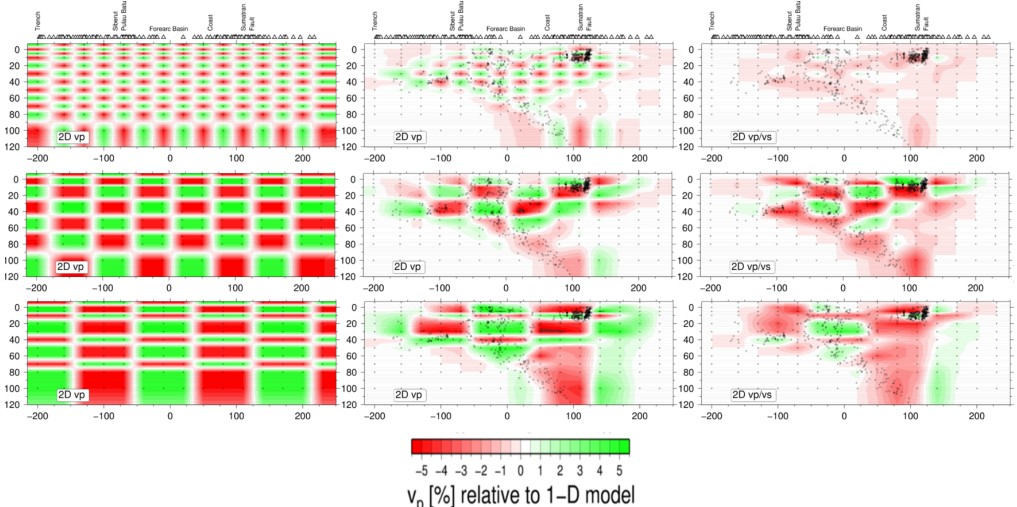

**Figure 6.** 2D synthetic checkerboard models with 5% velocity perturbation input anomalies (left column) and the inversion restoration for $v_p$ (centre column) and $v_p/v_s$ (right column) models. Crosses represent nodes used in the inversion and the reconstructions are plotted with the resulting hypocentre locations (black points). We calculated different checkerboard inversions using 1x1 and 2x2 (shown in the rows from top to bottom) grid node model perturbations. Noise was added to the synthetic data depending on the quality of the arrivals.

### 4.3.2 3D checkerboard test

For the 3D case we performed numerous checkerboard inversions using different checkerboard sizes. The checkerboard anomaly with 8 nodes (2x2x2 checkerboard, equivalent to 60x100x20 km) is reconstructed in the central part at depths between 5 and 50 km (Fig. 7). Below 50 km only the region beneath the volcanic arc shows sufficient ray coverage, but the profile view suggests vertical smearing below 50 km depth. In general, the resolution is good between the forearc islands and the SFZ between 5 and 50 km for the region above the Wadati-Benioff zone, so we will restrict our interpretation to this depth range. The shallow (<30 km) region beneath the eastern part of Siberut is characterized by aseismic behaviour during the deployment and the limited ray coverage results in insufficient recovery of the checkerboard in this region. A threshold for the spread values has been chosen to discriminate regions with high and low resolution and is superimposed on the resulting tomographic velocity models. The choice of threshold was carefully determined based on checkerboard tests, the ray coverage, and on the relative amplitudes of the spread values.




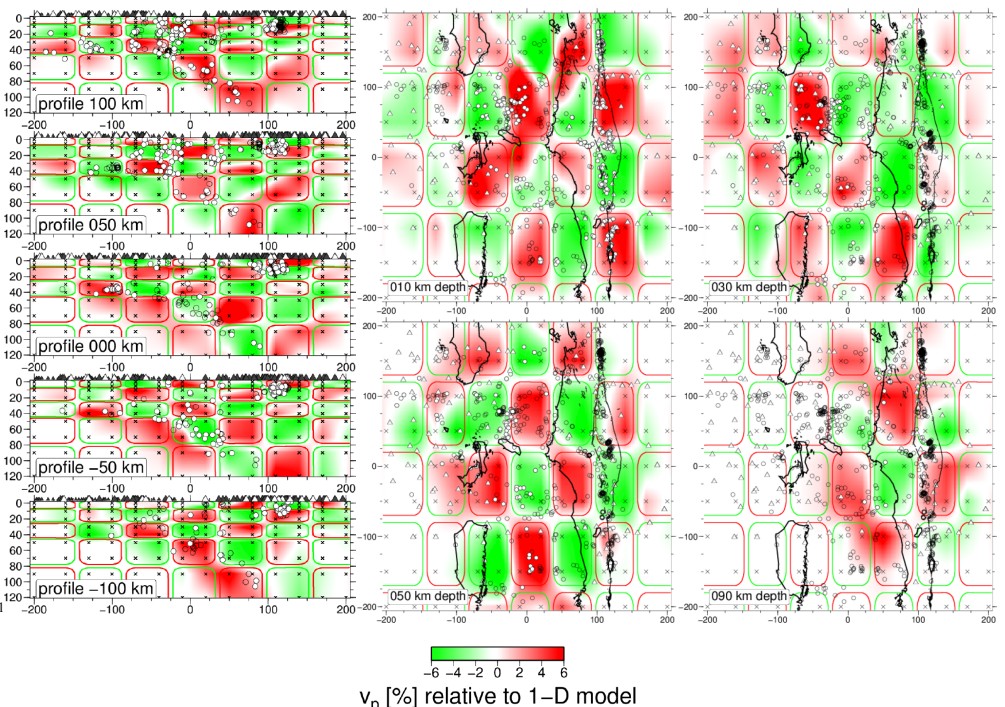

$v_p$ [%] relative to 1–D model

**Figure 7.** 3D synthetic checkerboard models with 5% velocity perturbation input anomalies and the inversion restoration for the 3D $v_p$ model. Other symbols as in Fig. 6.

### 4.3. 3D Synthetic restoration test

Restoring resolution tests were conducted to estimate the capacity of the data to resolve the geometry and amplitudes of potential velocity structures. We constructed synthetic $v_p$ velocity models with similar characteristics in amplitude and dimensions as the inversion results and further models with velocity anomalies representing the subducted IFZ. A possibly
10 modified crust along the IFZ was incorporated as obliquely oriented positive velocity anomalies at the expected position of the subducting crust (but with a larger thickness varying between 15 and 30 km). Further tests were conducted with shallow velocity anomalies. Figure 8 shows the 3D restoration of a synthetic model where we integrated different anomalies. The figure shows the restoration of an oblique velocity anomaly oriented in the direction of the subducted IFZ at 5 km depth and





at depths from the plate interface and trench-parallel velocity anomalies at 30 km depth. The absolute values of $v_p$ for the synthetic features are adequately reproduced (Fig. 8) in regions with good resolution as indicated by the spread value.

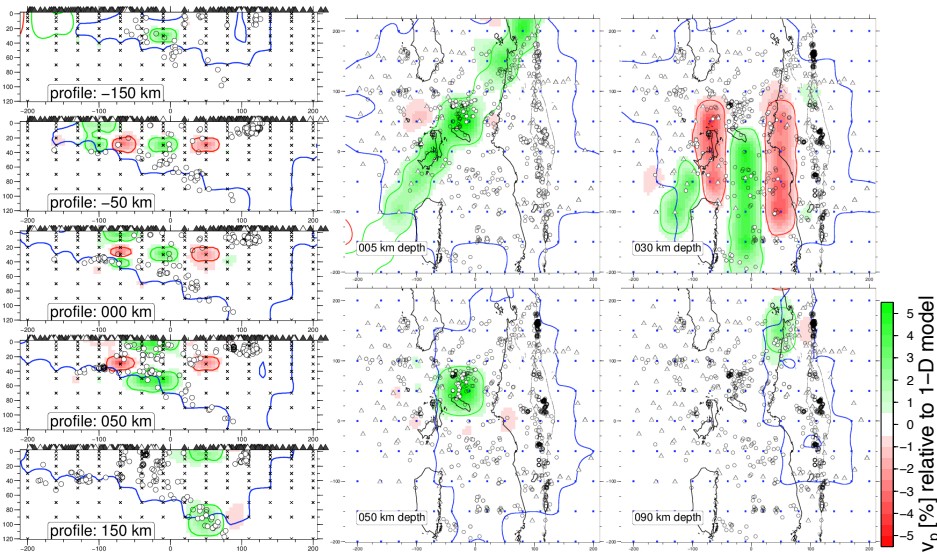

5  **Figure 8.** 3D synthetic models with 5% velocity $v_p$ perturbation input anomalies and the inversion restoration. The model consists of north-south trending anomalies (map view 30 km depth) and a NE-SW trending low velocity anomaly for both the shallow part of the model in 5 km as well as for the trace of the subducted IFZ. Red and green lines indicate the 5% contour lines of the input anomalies. Blue line encircles regions with good resolution defined by the spread value. Other symbols as in Fig. 6.

## 5 Results and Discussion

The 2D $v_p$ and $v_p/v_s$ velocity model is shown in Fig. 5, and the final 3D $v_p$ velocity model is shown in Figures 9 and 10, respectively. In the following, we discuss the main features for the different tectonic units, making use of the small-case labels in Figures 9 and 10.

15  ### 5.1 Accretionary Prism, Forearc Islands, and Forearc Basin

In the shallow part of the $v_p$ velocity model we observe regions of reduced $v_p$ velocities alternating with higher $v_p$ values at shallow depths (Fig. 5, ~10 km depth and Fig. 10, labels a,b and c). In the following, we discuss these regimes starting at the





trench and moving towards the mainland of Sumatra. The accretionary wedge composes the frontal prism adjacent to the deep-sea trench as well as the lower to middle continental slope seaward of the forearc islands. The accretionary domain (labelled a in Figures 9 and 10) is characterized by moderate velocities of ~5 km/s down to a depth of ~15 km, increasing to ~6 km/s above the landward dipping high-velocity zone (labelled f, Fig. 10). Velocities in the upper 15 km increase

underneath the forearc islands (labelled b) with values of ~6 km/s, which are also observed beneath the coast. The forearc basin between the islands and the mainland (labelled c) shows moderately low velocities of ~5 km/s down to 10 km depth. When considering the shallow forearc structure (<15 km), the trench perpendicular shallow structure variations are similar to the results of Collings et al. (2012) for the southern Mentawai Islands, in a way that the slow and fast domains alternate in the landward direction. The most obvious difference is that Collings et al. (2012) found low velocity values of approximately

5 km/s beneath the Mentawai forearc islands (Sipora, North and South Pagai), adjacent to faster material beneath the forearc basin. Our results image the region beneath the forearc islands as a trench-parallel (label b), elongated zone of increased velocities, sandwiched between the relatively lower velocities of the trenchward accretionary prism (a) and the landward forearc basin (c) the fast velocity anomalies below and between the islands might be interpreted as occurrence of faster accreted IFZ material beneath the Batu Islands.

The very shallow marine forearc at depths of 5 km is characterized by three regions of relatively reduced $v_p$ velocities of between 5 and 6 km/s. Faster regions (~6 km/s) are spatially related with the forearc islands Nias, Pulau Batu and Siberut and Pulau Pini (Fig. 9B). In-between the forearc islands the marine forearc is mostly characterized by $v_p$ velocities of 5 km/s. At depths of 20-30 km and 25 km east of the Mentawai fault, a trench parallel velocity anomaly of higher $v_p$ velocities (labelled d in Figures 9 and 10, indicative by the upwelling of contour lines) suggests a shallower location of the Moho

beneath the forearc basin and hence a reduced thickness of the overriding crust. Alternatively, this velocity anomaly might reflect a deep subducted seamount. Based on reflection data Singh et al. (2011) image an undulation of the top of the subducting slab in the Sumatran forearc to the south at 5°S and interpreted this as a subducted seamount. We exclude the possibility of a subducted seamount due to the size of the anomaly (200x80 km) and the absence of a similar feature in the seismicity (Fig. 1).

**5.2 Sumatran fault zone (SFZ) and volcanic arc**

While the offshore forearc is made up of young sediments from the Eocene to Holocene, the mainland shows a ~130 km wide belt of different rock units along the SFZ. The SFZ is characterized by high seismicity rates (e.g. Weller et al., 2012) due to stress and strain partitioning from the oblique subduction (McCaffrey et al., 2012). This belt is mostly composed of Permian to Jurassic sedimentary rocks, Eocene volcanic rocks and Jurassic to Eocene intrusive units (Crow und Barber,

2005). The 3D velocity model along the SFZ is characterized by only minor changes of $v_p$ along strike. Seismic velocities of 7.8 km/s (indicative of continental Moho) are reached at depths larger than 30 km and outside the region of good resolution. A Moho depth between 28 to 40 km is in-line with Moho depths from receiver functions in the region of the caldera of lake



Toba (Fig. 1) (Sakaguchi et al., 2006, Kieling et al., 2011) and similar to the Moho depths inferred from receiver functions (Gunawan et al., 2011).

$V_p/v_s$ values beneath the SFZ *(depths≤20 km)* are between 1.65 and 1.72 (Fig. 5) and similar to the minimum 1D velocity model of Weller et al. (2012), which used the same stations to derive an optimum 1D model for the SFZ region only. These

low $v_p/v_s$ ratios seem to be characteristic for the shallow crust in the region of the SFZ. Muksin et al. (2013) conducted a local earthquake tomography for the shallow crust (<15 km) at 2°N and find similar lower $v_p/v_s$ values away from the SFZ. Equally, Koulakov et al. (2009) image predominantly lower $v_p/v_s$ ratios below 1.8 for the region 100 km northwest of our study area (Tb in Fig. 1). Our findings differ from the velocity model of Koulakov et al. (2009, 2016), in that we find only weak indications of a patchy low velocity zone beneath the magmatic arc at 30 km depth only.

**5.3 Subducting Oceanic Lithosphere**

Where the slab is still in contact with the overriding plate, the oceanic Moho is imaged as the inclined 7.8 km/s $v_p$ contour line (Fig. 5, panel C and Fig. 10, label f). The plate interface, inferred from seismicity, is located at approximately 25 km depth below the forearc islands (Fig. 5), a little deeper than beneath the Pagai Islands at 3°S, where it was found at 20 km depth (Collings et al., 2012), but significantly deeper than the plate interface from seismicity and refraction seismic found at

15 km depth beneath Simeulue Island at 2.5°N (Tilmann et al., 2010, Shulgin et al., 2013).

Seismicity 25 km west of Nias (Fig. 2) is part of a coast-parallel band of seismicity. This band of high seismicity corresponds to the transition between regions of significant coseismic (downdip) and aseismic slip (updip) of the 2005 earthquake (Hsu et al., 2006) and extends northwestwards until Simeulue Island, roughly following the 500 m isobath

contour lines (Tilmann et al., 2010). The depth variations of seismicity along this seismicity band suggest that the seismicity transition from aseismic to seismic behaviour in downdip direction (Lange et al., 2007, Tilmann et al., 2010, Lange et al., 2010) might not be controlled by depth and hence lithostatic pressure.

The inclination of the subducting plate is approximately 25° within the depth range between 40 and 80 km, also based on the

seismicity, as the resolution and grid spacing is insufficient for imaging subducting oceanic crust. There are hints of the contrast between the subducting high velocity slab and the mantle wedge in the form of a dipping velocity contour (e.g. Fig 10D), but it is only imaged in a patchy way at the limit of the resolved area. At larger depths, seismicity can be traced down to 220 km with an inclination of approximately 36° (Lange et al., 2010) but the velocity structure is no longer resolved (Fig. 9E).


Fig. 9F shows a section through the 3D $v_p$ velocity model following the plate interface (defined by the SLAB1.0 model, Hayes et al., 2012). The dominant feature in this panel is the contrast between crust and mantle, allowing us to identify the position of the toe of the mantle wedge just landward of the forearc islands (except Pulau Pini, which is already well above



the mantle wedge). No obvious along-strike change can be identified in the mantle wedge. In particular, the velocity model does not reveal indications of velocity anomalies in the direction of the subducted IFZ, although the trace of the subducted IFZ is reflected by seismicity down to ~200 km depths (Fauzi et al., 1996; Lange et al. 2010); in Fig. 9F it is visible as a band of seismicity striking north (i.e. to top right in the figure) from Pulau Batu. The synthetic restoration tests (Fig. 8)

5   document that the inversion is capable to resolve a ~40 km wide velocity anomaly, if there would be any. Considering such large scale structures we conclude that the subducted IFZ did not disturb the velocity structure at depths of the plate interface, e.g. by releasing fluids and enhancing melt production but clearly had a significant impact on the rheological conditions within the slab since it enhances intermediate depth seismicity down to large depths (Lange et al., 2010). Some of the events, labelled with f in Fig. 10, panel C are located 10-15 km below the plate interface defined by the global slab model

10  (Hayes et al., 2012). Based on their hypocentral depths we interpret them as being spatially related to the oceanic crust to mantle transition (e.g. near the oceanic Moho) or even possibly occurring in the uppermost oceanic mantle. For the North Chilean subduction zone, Bloch et al. (2014) found a similar group of events ~8 km below the plate interface for the northern Chilean subduction zone and in depths between 30 and 60 km and proposed them to be spatially related to the oceanic Moho.

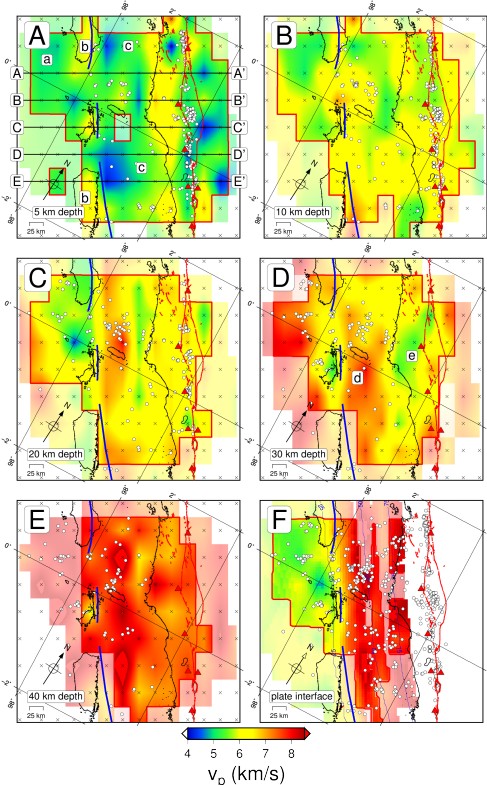

**Figure 9.** 3D $v_p$ model: Depth sections (panels A-E) and curved section along the plate interface as defined by the global SLAB1.0 model (Hayes et al., 2012) (panel F). Red line encircles regions of good resolution defined by the spread value. White circles indicate events within 10 km of the section depth, except panel F, where all events used for the inversion are shown. Volcanoes (Smithsonian Institute) shown with red triangles. The Mentawai Fault (blue line offshore) and the Sumatran Fault (red line onshore) are also shown. See text for explanation of characters. Other symbols as in Fig. 7.



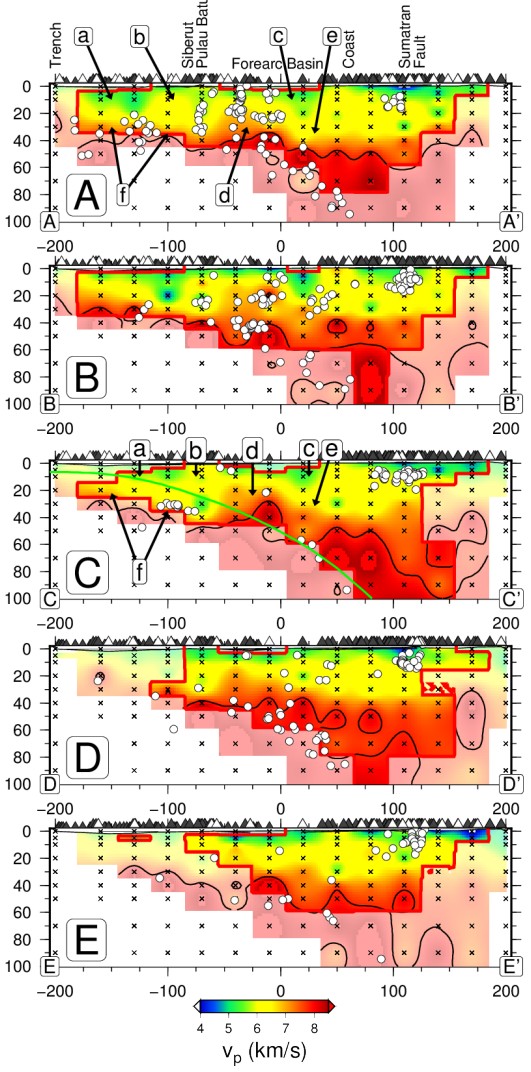

**Figure 10.** Cross-sections along trench-perpendicular trending profiles through the 3D $v_p$ model. See Fig. 9, panel A for location of cross sections. White circles indicate events within 10 km of the profile and stations closer than 25 km to the profile are shown by white triangles, the remaining ones by black triangles. The 46 OBS stations of the 2-week deployment are shown with smaller triangles. Grid nodes are shown with crosses. Red line encircles regions of good resolution defined by the spread value. Green line in panel C indicates the plate interface as defined by the global SLAB1.0 model (Hayes et al., 2012). 7.8 km/s $v_p$ contour line is indicated by a black line. See text for explanation of characters. Other symbols as in Fig. 7. Note that the geographic labels at the top refer to all profiles, but that only profile C (Batu Islands) and E (Siberut) actually cross a forearc island.




### 5.4 vp/vs model of the foearc

In our study region around Nias and Siberut, we find mostly only moderately elevated $v_p/v_s$ values (up to 1.85, Fig. 5B), whereas Collings et al., 2012 find strongly elevated $v_p/v_s$ values (up to 2.0) down to the plate interface below the Pagai

Islands. The largest values are found west of the forearc basins in the region of the Mentawai fault just landward of the forearc islands. Since rays of the 2D $v_p/v_s$ velocity model mostly sample the region northeast of Pulau Batu (Fig. 2) this likely reflects a local $v_p/v_s$ anomaly close to the equator rather than being a feature present along the whole along-strike length of the study-region. The reason for this region of elevated $v_p/v_s$ remains enigmatic. Possible explanations include fluids related to pathways created by the Mentawai fault or structural differences due to subducted material from the IFZ.

Although $v_p/v_s$ ratios are moderately elevated (up to 1.85) we cannot identify large-scale alteration of the mantle wedge due to surplus liquids from a strongly hydrated IFZ because serpentinized material is characterised by clearly elevated $v_p/v_s$ and reduced $v_p$ values (e.g. Carlson & Miller, 2003). Because mantle serpentinisation favours aseismic sliding and is related to the downdip extent of the seismogenic zone (e.g. Hyndman et al. (1997), Oleskevich et al., (1999), the lack of large scale serpentinisation could explain why the seismogenic plate interface extends into the forearc mantle off Sumatra (e.g. Simoes

et al., 2004, Collings et al., 2012). In particular, the stalling of the 2005 rupture, which was suggested to be limited by the subducted IFZ (Chlieh et al., 2008), might be related to rheological properties and heterogeneities along the plate interface. Based on MCS data, Henstock et al. (2016) identify an isolated 3 km basement high close to the 2005 slip termination as well as along-strike variations of basement relief. Such features are large enough to affect the rheological behaviour of the plate interface such as coupling but below the resolution of our local earthquake tomography.

## 6 Conclusions

We present 2D and 3D velocity models from a local earthquake tomography using data from a dense network of seismic stations covering the onshore and offshore domain of the northern Sumatra forearc close to the equator. The models resolve the structure of the forearc including the accretionary prism, forearc islands, and the forearc basin, the mantle wedge and the volcanic arc down to a maximum depth of ~60 km. The downgoing slab is traced by inclined velocity contour lines at depths

<40 km. The oceanic crust has a velocity of ~7 km/s and is located at a depth of ~25 km beneath the forearc islands (based on the seismicity depth distribution). $V_p$ velocities beneath the magmatic arc, which spatially coincides with the SFZ, are around 5 km/s at 10 km depth and the $v_p/v_s$ ratios in the uppermost 10 km are low, indicating the presence of felsic lithologies typical for continental crust.

The forearc basins west and east of the Mentawai Islands are characterized by velocities of ~5 km/s down to 15 km depth. Although the region is characterized by the subducted IFZ, which influences seismicity down to depths of 200 km, the 3D velocity model at depths of the plate interface shows prevailing trench parallel structures suggesting that the subducted IFZ

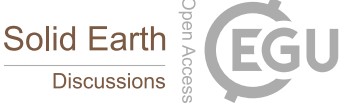

did not significantly modify the velocity structure at seismogenic depths. At very shallow depths (~5 km) and below the forearc islands (Pulau Batu, Siberut, Nias) higher $v_p$ velocities of ~6 km/s are found.

In depths of 20-30 km and ~25 km east of the Mentawai fault, a trench-parallel velocity anomaly of higher $v_p$ velocities

might suggests a shallower location of the Moho beneath the forearc basin and hence a reduced thickness of the overriding crust.

Elevated $v_p/v_s$ ratios of 1.85 are found in the overriding crust in the region of the Mentawai fault, which might be related to fluids. However, $v_p/v_s$ ratios are still too small to support a large-scale serpentinisation of the continental mantle and could

explain why the seismogenic plate interface (observed as a locked zone from geodetic data) extends below the continental forearc Moho in Sumatra.

## Data availability

Seis-UK data are available from IRIS (www.iris.edu) using the network code ZB (2007-2009). https://geofon.gfz-potsdam.de/Network codes GE, and 7A (2008) are stored at GEOFON data centre (https://geofon.gfz-potsdam.de/). GFZ

instruments were provided by the GIPP. The data from the permanent Indonesian network (network code IA) are stored at BMKG (www.bmkg.gov.id).

## Author contribution

DL, FT, TH, AR, DH were involved in the installation of the seismological stations. FT, TH, AR and DH designed the experiment. AR and DH supervised the project. DL processed the data, conducted the inversions and prepared the artwork.

DL led the development of the manuscript supported by significant contributions from FT, TH and HK who contributed to the ideas, concepts and interpretation presented in this manuscript.

## Competing financial interests

The authors declare no competing financial interests.

## Acknowledgments,

We thank the SeisUK facility in Leicester for the loan of the instruments and the logistic support during this project, loan 828 (Brisbourne, 2012). We acknowledge the support of the colleagues at Geotek-LIPI for this project. LIPI-EOS let us share some of the sites of the SuGaR GPS network. We thank the captain and crew of the Andalas for excellent work in the




field. Furthermore, we thank the master and crew of R/V SONNE cruises SO-198 and SO-200 for the deployment and recovery of the OBS. OBS instruments were provided by OBIF. We thank Lisa C. McNeill and Penny Barton who participated in the acquisition of the OPS data. The project was funded by NERC (NE/D00359/1). EOS (Earth Observatory of Singapore) is thanked for supporting logistical costs of deployment on Mentawai and Batu Islands. We thank the

Indonesian BMKG and German GEOFON for the station data from their permanent networks. Furthermore, we are indebted to all field crews for their excellent work under tropical conditions. We thank Imam Suprihanto, Bambang Suwargadi and Rachel Collings for support during the fieldwork. Finally, we gratefully acknowledge the cooperation of many Sumatran landowners, communities, and institutions for support and for allowing us to install the seismic stations on their property.

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
