# Peer review of "Structure of the Central Sumatran Subduction Zone Revealed by Local Earthquake Travel Time Tomography Using Amphibious Data"

_Solid Earth, 2017_

## Short Comment (SC1) · 9 Jan 2018

The excellent manuscript by Lange and others describes a local Vp and Vp/Vs tomography of the Mentawai seismic gap region in Sumatra, Indonesia based on an amphibious seismic array. The paper lays out the methodology very clearly, showing the potential and limitation of the data through a series of resolution and synthetic tests. The methodology involves 1D, 2D, and 3D velocity models. The analysis reveals the seismic structure of the region down to about 60 km depth. Of particular interest is the velocity structure down-dip the megathrust that highlights the position of the toe of the mantle wedge, a region thought to control the down-dip extent of earthquakes.

[Figure]

Sumatra stands out because there is no clear long-term nor short-term slow slip events there, while it is found pretty much in every other subduction zone when data is available. This excellent paper is ready for publication as is, except for a few typos (misuse of subjunctive at line 5 on page 16, typo in section title 5.4). Figures 9 and 10 need to be bigger.

---

## Referee Comment (RC1) · I.Yu. Koulakov (Referee) · 19 Jan 2018

This study deals with a very interesting region of Central Sumatran subduction zone. In some previous studies (e.g. Koulakov et al., 2016), it was proposed that the repeated supereruptions of Toba were controlled by the subduction of the Investigator Fault Zone (IFZ) that separate two plate segments of different ages and possibly brings to the mantle an anomalous amount of water. The topography of the forearc along the IFZ line behave differently than in other subduction segments along the Sumatra coast that probably indicates that the IFZ greatly controls the accretion process. I expected that the IFZ should be the most prominent structure in the area considered in this study,

and I am a little bit disappointed that IFZ-related structures are almost not revealed in the tomography results. I think the authors should pay more attention to this problem.

From the methodological point of view, this paper is an excellent example of the SIMULP-based description similar to dozens of previously published papers based on this tomography code. All the steps of the traditional SIMILP workflow have been carefully completed and described. The problem is that many statements taken as an axiom by the SIMULP users seem to me not grounded and adequate. The criticism presented below relates to all SIMULP-based studies, not to this particular case. Therefore it would be unfair from my side to insist on changing everything in this specific paper. However, I would be happy if some of my arguments will be taken into account during the revision and will be useful in future studies.

The major problem of the algorithm is defining the parameterization grid according to the expected resolution, so that the grid spacing is equal to the size of minimum resolved anomaly. This is a completely wrong strategy. If the size of anomaly is compatible with the grid spacing, such anomaly would appear completely different if its center coincides with one node or it is located between nodes. In this case, the solution will be grid dependent, which is a serious flaw of tomography. Such one-node-based anomalies will be completely changed if, for example, you shift the grid a half step. To avoid such grid dependency, we should define the grid spacing much smaller than the size of anomalies, so that every resolved anomaly is based on several nodes. The stability of the inversion should be controlled not by grid spacing, but by smoothing and regularization in inversion. We can see such grid-dependency in the results presented in this paper. For example, in the Vp/Vs ratio section in Figure 5, we see that at X=-100 km, there is shallow blue and deep red; in the next column at X=-70, there is shallow red and deep neutral; then at X=-40, there is heavy red anomaly etc. It is clear, if the points were shifted to half step and installed at -85, -55, -25, the anomalies would be completely different.

Another problem of the SIMULP workflow is using the trade-off curve for estimating

optimal damping parameters. This curve is calculated from a series of inversions with different damping values in the first iteration. Why should it be valid for the inversions in multiple iterations? It is clear that number of iterations also affect the stability of the inversion and, therefore, connected with damping. For example, a fixed damping may provide an overdamped solution in one iteration and underdamped solution after 10 iterations. It is obvious that an optimal damping value estimated from the L-curve for one iteration is not optimal for ten iterations. In addition, I have never seen any study supported by modeling results that confirmed that the value in the corner of the L-shaped trade-off curve does really provide the best damping. At the same time, I know opposite examples showing that the best damping values may be far from the corner point.

I have serious concerns about performing synthetic modeling. The good synthetic modeling should provide the realistic assessment for the resolution capacity and, therefore, it should adequately simulate the real workflow that is used in case of processing of experimental data. In passive source tomography, the most difficult problem is the trade-off between source locations and velocity model. For example, if a source is located between positive and negative velocity anomalies, the initial step of source location in the 1D velocity model would shift the coordinates and origin time so that the residuals would be close to zero. In turn, it will make problematic recovering the velocity model. It is clear that if we start recovering of synthetic model from the step of source location in the 1D starting model (as we do for the experimental data), the result would appear not as nice as in the case when we use the residuals directly calculated from synthetic model. Similar difficulties take place in the case of deep sources. Shifts of source coordinates and origin times "kill" any residuals that would allow us to restore layered structures, such as in the lower panels in Figure 6. The problem of the SIMULP workflow is that in synthetic modeling, they start restoring anomaly without performing the step of initial source locations. The residuals directly computed from the synthetic models provide very nice restoration of anomalies. However, such modeling is not related to realistic resolution capacity, which is strongly perturbed by the

trade-off between the source and velocity parameters.

Another problem of synthetic modeling in the SIMULP workflow is that the anomalies are predefined in the same nodes as used for inversion. Successful restoring the anomalies centered in the nodes with spacing of 30x10 km gives an impression that the existing observation scheme would allow us to resolve such size of anomalies in the case of experimental data. Obviously, it might be true only if the anomalies perfectly centered with nodes. However, if an anomaly of 30x10 km size is located between nodes, it would obviously not be recovered, or strongly smeared. As the locations of real anomalies in the nature are not known, such modeling gives wrong assessment for the resolution. The shapes of synthetic models should be completely independent of the parameterization grid. In this case, we will recover not only amplitudes of anomalies (as in the present case), but also their locations and shapes, that is much more complicated.

Other specific comment on the paper.

I did not understand the meaning of the 2D modeling performed in this study. Was it based on all data in the area? Does it mean that velocity along Y-coordinate is presumed to be constant? If yes, I would hardly expect any stable solution because of the existence of significant heterogeneities along the trench line (for example, due to the presence of IFZ).

Why the Vp/Vs ratio is only shown for the 2D model, and not for the 3D model? It would be interesting to see the variations of the Vp/Vs ratio in the map view. I expect that if the IFZ is saturated by water, it would be seen in the Vp/Vs distributions.

In horizontal sections, it would be better to present relative anomalies instead of absolute velocity. In the present case, it is hard to see any nuances in dominantly green, yellow or red colors corresponding to absolute velocities at specific depths. The traditional "blue-white-red" scheme would provide much clearer images for the velocity variations.

In vertical sections, in addition to absolute velocity, it would be helpful to present also the relative anomalies. In some cases, they appear to be very informative for interpretation.

P8L3: Does the 76% of reduction correspond to absolute residuals (L1) or squared values (L2)?

Figure 6 is mentioned in the text prior Figure 5.

Was there any 3D synthetic modeling for the Vp/Vs ratio? Why the synthetic recovery results in the 3D case are shown only for the Vp, and not for the Vp/Vs?

In Figure 8, the contours of the initial synthetic model should be highlighted.

---

## Referee Comment (RC2) · Anonymous Referee #2 · 29 Jan 2018

This is an excellent study of the velocity structure of a significant section of the Suma-tran subduction zone including a transition from strong to low coupling. The inclusion of offshore and onshore seismic stations provides good imaging capability, and the tomography is carried out in a careful manner to produce reliable results. They show basins and variation in the lower crustal structure of the overlying plate.

Specific comments focus on aspects of the interpretation:

1. P 2 geodetic discussion. Add more discussion of Chlieh 2008 heterogeneous cou-pling, including some simple line (maybe 0.4 coupling) on Figures 1-2. Then the 3D velocity for the upper plate can be compared to coupling heterogeneity, which varies

from 0 to 1 in the study area.

2. P14, L13-14 Add more discussion and consideration of the process of accretion, and how the IFZ and other fracture zones could influence. With the difference in angle between the IFZ and the trench-megathrust, and the partitioned slip, it seems that for a specific forearc plate-interface depth, the location of the IFZ would move southeast with time as the subducted plate descends, and thus could create margin-parallel accreted features.

3. P14, L18-24 and conclusions. High-velocity feature d (Fig 10) is interesting in that it seems correlated with the region of low coupling. The 3D Vp shows that it is a spatially distinct feature. Is there actually some localized process of crustal thinning at that location? Since the forearc crust is considered to be assembled through accretion, it seems more likely that it is mafic block that has been accreted.

4. P18 and conclusions, Vp/Vs. If the high Vp/Vs is related to the Mentawai fault back-thrust, then it might be a zone of releasing and transporting subduction fluid into and through the upper plate crust. If the crust is permeable, then the subduction fluid would not be trapped in the mantle wedge corner and thus the lack of serpentinization.

Technical corrections:

5. P2, L20 change 'is undebated' to 'unabated'

6. P2, L27-29. Add specific depths for the Moho summary.

7. Figure 1. Green for Mentawai is poor color choice because same as land. Add box for area of Figures 2, 9

8. P4, L20. Change 'targets' to 'target'

9. Figure 4 caption, P11 L16, and elsewhere. Put in the specific spread cutoff value.

10. P10, L2. Remove '(Fig. 5)' which is not the checkerboard.

11. Figure 9. Make this bigger so that it takes up the full page. It is too hard to see.

12. Figure 10. Use the same Vp color scale for this and for Figure 5 so that readers can compare. Mark the Mentawai fault.

---

## Author Comment (AC1) · 6 Mar 2018

Dear Dr. Barbot,

We thank you for your careful and constructive comments on our paper.
We incorporated all your suggestions.

Best wishes,
Dietrich Lange  co-authors
* * *

---

## Author Comment (AC3) · 6 Mar 2018

Dear anonymous reviewer,

We thank you for your careful and constructive review. In the accompanying revision notes below we listed in detail on how we incorporated your helpful comments and suggestions. Your original review text is shown in black and our answers in blue colour.

Yours sincerely,
Dietrich Lange and co-authors

[Figure]

This is an excellent study of the velocity structure of a significant section of the Sumatran subduction zone including a transition from strong to low coupling. The inclusion of offshore and onshore seismic stations provides good imaging capability, and the tomography is carried out in a careful manner to produce reliable results. They show basins and variation in the lower crustal structure of the overlying plate.

Specific comments focus on aspects of the interpretation:

1. P 2 geodetic discussion. Add more discussion of Chlieh 2008 heterogeneous coupling, including some simple line (maybe 0.4 coupling) on Figures 1-2. Then the 3D velocity for the upper plate can be compared to coupling heterogeneity, which varies from 0 to 1 in the study area.

We superimposed the plate coupling (Chlieh et al., 2008) with red contours onto the ray coverage (Figure 2).

We modified the sentence on page 2:
*"Sieh et al. (2008) estimate the slip deficit below Siberut Island since the large ruptures of 1797 and 1833 to be 8 m and a reduced slip deficit of 5 m for the Batu Islands due to the lower degree of coupling in the region of the Batu Islands (Fig. 2 and Chlieh et al, 2008)."*

2. P14, L13-14 Add more discussion and consideration of the process of accretion, and how the IFZ and other fracture zones could influence. With the difference in angle between the IFZ and the trench-megathrust, and the partitioned slip, it seems that for a specific forearc plate-interface depth, the location of the IFZ would move southeast with time as the subducted plate descends, and thus could create margin-parallel accreted features.

We agree and added to the discussion of P14:

*"On geological time scales the intersection of the IFZ with the marine forearc migrates southeast as the subducted plate descends, and thus might have created margin-parallel accreted features north the current intersection of the IFZ with the trench (e.g. north of Siberut Islands). However, we cannot find significant along-strike variations of vp between the Mentawai Islands and the trench (e.g. labelled a in Fig. 10 and 11) which might equally be explained by accretion of seamounts (Fig. 1, 99.5° E/4.5° S)."*

3. P14, L18-24 and conclusions. High-velocity feature d (Fig 10) is interesting in that it seems correlated with the region of low coupling. The 3D Vp shows that it is a spatially distinct feature. Is there actually some localized process of crustal thinning at that location? Since the forearc crust is considered to be assembled through accretion, it seems more likely that it is mafic block that has been accreted.

We completely agree that this might serve as an alternative explanation and added this sentence:
*"Alternatively, this trench parallel velocity anomaly of higher vp velocities (labelled d) might be explained by an accreted mafic block."*

4. P18 and conclusions, Vp/Vs. If the high Vp/Vs is related to the Mentawai fault backthrust, then it might be a zone of releasing and transporting subduction fluid into and through the upper plate crust. If the crust is permeable, then the subduction fluid would not be trapped in the mantle wedge corner and thus the lack of serpentinization.

Technical corrections:

5. P2, L20 change 'is undebated' to 'unabated'

Corrected.

6. P2, L27-29. Add specific depths for the Moho summary.

We added the specific Moho depths:
*". . . . . . on gravity surveys and wide-angle refraction and local earthquake tomography (Siberut: Simoes et al. 2004, Kieckhefer et al. 1980, 30 km Moho depth; Aceh basin*

*and Simeulue: Dessa et al. 2009, Klingelhoefer et al. 2010, Tilmann et al. 2010, 21-25 km Moho depth; Southern Mentawai Islands: Collings et al. 2012, less than 30 km Moho depth)."*

7. Figure 1. Green for Mentawai is poor color choice because same as land. Add box for area of Figures 2, 9.

We now show the Mentawai fault in yellow colour and added two boxes for the locations of Figure 2:
*"Black boxes indicate locations of Figures 2,6,7 and 9."*

8. P4, L20. Change 'targets' to 'target'

Done.

9. Figure 4 caption, P11 L16, and elsewhere. Put in the specific spread cutoff value.

We added to the caption of Figure 4 (2D spread values):
*"Cut-off spread values are 2.1 and 1.9 for vp and vp/vs, respectively."*
and we modified the captions of Figures 9  10:
*"Red line encircles regions of good resolution defined by a cut-off spread value of 1.5."*
Additionally, we labelled the contours with the cut-off spread (Figure 8).

9a.: P10, L2. Remove '(Fig. 5)' which is not the checkerboard.

Done.  Previous Figure 5 is now Figure 8 and Figures 8,7 and 6 were renamed to Figures 7,6 and 5.

11. Figure 9. Make this bigger so that it takes up the full page. It is too hard to see.

We enlarged Figure 9 onto one page.

12. Figure 10. Use the same Vp color scale for this and for Figure 5 so that readers can compare. Mark the Mentawai fault.

We changed the colour scale in Figure 8 (Previous Figure 5). Now the 2D and 3D vp

models show the same colour scale. We marked the Mentawai fault in Figure 8 and 10.

---

## Author Comment (AC4) · 6 Mar 2018

Dear Dr. Koulakov,

We thank you for your careful and constructive review. In the accompanying revision notes below we listed in detail on how we incorporated your comments and suggestions. Your original review text is shown in black and our answers in blue colour.

Yours sincerely,
Dietrich Lange and co-authors

[Figure]

This study deals with a very interesting region of Central Sumatran subduction zone. In some previous studies (e.g. Koulakov et al., 2016), it was proposed that the repeated supereruptions of Toba were controlled by the subduction of the Investigator Fault Zone (IFZ) that separate two plate segments of different ages and possibly brings to the mantle an anomalous amount of water. The topography of the forearc along the IFZ line behave differently than in other subduction segments along the Sumatra coast that probably indicates that the IFZ greatly controls the accretion process. I expected that the IFZ should be the most prominent structure in the area considered in this study, and I am a little bit disappointed that IFZ-related structures are almost not revealed in the tomography results. I think the authors should pay more attention to this problem.

Since a fracture zone is characterized by oceanic material on both sides of the fracture the expected density contrast and hence velocity contrast is small. From a conceptual standpoint the most significant difference in velocity contrast originates from different oceanic plate ages on both sides of the fracture zone. Regarding the seamounts found on the IFZ very little is known about their deeper velocity structure related to the IFZ. We already state in the text (Page 19) that for smaller scales there are along-strike variations known for the Sumatran margin:

*"Based on MCS data, Henstock et al. (2016) identify an isolated 3 km basement high close to the 2005 slip termination as well as along-strike variations of basement relief. Such features are large enough to affect the rheological behaviour of the plate interface such as coupling but are below the resolution of our local earthquake tomography."*

From the methodological point of view, this paper is an excellent example of the SIMULP-based description similar to dozens of previously published papers based on this tomography code. All the steps of the traditional SIMILP workflow have been carefully completed and described. The problem is that many statements taken as an axiom

by the SIMULP users seem to me not grounded and adequate. The criticism presented below relates to all SIMULP-based studies, not to this particular case. Therefore it would be unfair from my side to insist on changing everything in this specific paper. However, I would be happy if some of my arguments will be taken into account during the revision and will be useful in future studies.

We feel that these fundamental points risen by the reviewer might be a bit out of scope for our paper. As you already mention SIMULPS/SIMUL2000 is a very established inversion code used for more than two decades for many local earthquake inversions across different scales and virtually all geological settings. For many of these regions an independent velocity model was known from active seismics which SIMUL2000 could reconstruct, obviously with lower resolution than the reflection seismic data.

The major problem of the algorithm is defining the parameterization grid according to the expected resolution, so that the grid spacing is equal to the size of minimum resolved anomaly. This is a completely wrong strategy. If the size of anomaly is compatible with the grid spacing, such anomaly would appear completely different if its center coincides with one node or it is located between nodes.

In this case, the solution will be grid dependent, which is a serious flaw of tomography. Such one-node-based anomalies will be completely changed if, for example, you shift the grid a half step. To avoid such grid dependency, we should define the grid spacing much smaller than the size of anomalies, so that every resolved anomaly is based on several nodes. The stability of the inversion should be controlled not by grid spacing, but by smoothing and regularization in inversion. We can see such grid-dependency in the results presented in this paper. For example, in the Vp/Vs ratio section in Figure 5, we see that at X=-100 km, there is shallow blue and deep red; in the next column at X=-70, there is shallow red and deep neutral; then at X=-40, there is heavy red anomaly etc. It is clear, if the points were shifted to half step and installed at -85, -55, -25, the anomalies would be completely different.

We disagree that this would be completely different. In case we would shift the grid spacing for the 2D vp/vs inversion by 50% of the node spacing the very localized anomaly (based on the nodes at 0 and 5 km depth) at Figure 8 (previously Figure 5), panel B x=-100, Y=-5 would still reveal reduced vp/vs values, but might be a bit more smeared out between two nodes since the velocity between the grid nodes in SIMUL2000 is estimated from spline interpolation. We cannot follow the statement that the grid spacing is too coarse since every significant anomaly in Figure 8 is based on more than 2 nodes. Of course, SIMUL2000 does not allow to change the grid spacing during the inversion steps. However, we tried different grid spacings (and for the 2D model different azimuths of the profile) to assure that the inversion results are robust. In fact, we could not find significant differences of spatial distribution of anomalies within the resolution given by the checkerboard test (e.g. 2x2 nodes for the 2D velocity model, equals 30 x 10 km for the central part of the velocity model, Figure 5)

One of the inherit challenges of local earthquake tomography is the spatially heterogeneous resolution originating from the heterogeneous ray coverage. The suggested much smaller grid spacing compared to the size of the anomalies, let's say 5 km grid spacing, would allow an apparently higher resolution. However, such a small grid spacing enhances the danger of oscillating anomalies, all of which might not be resolved. Therefore we prefer to continue with an approach where we use in the final paper a grid spacing which is about two times than the best expected resolution. During the data processing we tested from sparse to fine grid spacing and carefully choose the final grid spacing based on our expected resolution.

We added the word carefully to the main manuscript in the method section:
*"After carefully testing different spacing parameters for 2D and 3D inversions in all three directions, we selected the node spacing as a compromise between resolution and stability of the inversion. "*

Another problem of the SIMULP workflow is using the trade-off curve for estimating optimal damping parameters. This curve is calculated from a series of inversions with
different damping values in the first iteration. Why should it be valid for the inversions in multiple iterations? It is clear that number of iterations also affect the stability of the inversion and, therefore, connected with damping. For example, a fixed damping may provide an overdamped solution in one iteration and underdamped solution after 10 iterations. It is obvious that an optimal damping value estimated from the L-curve for one iteration is not optimal for ten iterations. In addition, I have never seen any study supported by modeling results that confirmed that the value in the corner of the L-shaped trade-off curve does really provide the best damping. At the same time, I know opposite examples showing that the best damping values may be far from the corner point.

We follow the philosophy that the best model in local earthquake tomography is:
a.) most simple ("smooth") model
b.) the model that explains the data best
and exactly this is the physical reason for the damping curve. A damping value far away from the corner point does not match both criteria. We admit that the data and model variance is just taken from the first inversion and SIMUL2000 just uses one damping value for all inversions. If the changes in the first iteration are, for example, only about one tenth of the total, then the concerns would be somewhat be justified. However, for the inversion runs the first iteration was always much larger than in subsequent iterations, and changes getting progressively and quickly smaller. Therefore the first iteration step is representative of the total damping because it includes the largest changes.

We added a sentence to the manuscript to clarify how exactly the damping value was chosen and how the velocity models relate to the choice of damping value:

*"SIMUL2000 uses one damping value for all inversion steps and the model and data variance for the trade-off curve is taken from the first inversion step. We made various inversions with different damping values and found that the spatial distribution of anomalies stays similar, but with varying amplitudes of the anomalies. "*

This stable behaviour of the inversion we relate to the choice of grid spacing (see paragraph above), which prevents the occurrence of oscillations in the spatial domain.

I have serious concerns about performing synthetic modeling. The good synthetic modeling should provide the realistic assessment for the resolution capacity and, therefore, it should adequately simulate the real workflow that is used in case of processing of experimental data. In passive source tomography, the most difficult problem is the trade-off between source locations and velocity model. For example, if a source is located between positive and negative velocity anomalies, the initial step of source location in the 1D velocity model would shift the coordinates and origin time so that the residuals would be close to zero. In turn, it will make problematic recovering the velocity model. It is clear that if we start recovering of synthetic model from the step of source location in the 1D starting model (as we do for the experimental data), the result would appear not as nice as in the case when we use the residuals directly calculated from synthetic model. Similar difficulties take place in the case of deep sources. Shifts of source coordinates and origin times "kill" any residuals that would allow us to restore layered structures, such as in the lower panels in Figure 6. The problem of the SIMULP workflow is that in synthetic modeling, they start restoring anomaly without performing the step of initial source locations. The residuals directly computed from the synthetic models provide very nice restoration of anomalies. However, such modeling is not related to realistic resolution capacity, which is strongly perturbed by the trade-off between the source and velocity parameters.

It is not correct that the events are not located prior to the inversion in the synthetic test. Please see the SIMULPS manual (Evans et al., 1994), page 13, paragraph *nitmax*, where it is stated:
"Events input as "earthquakes" (Unit 04) are first relocated, which may not be what you intended"

We re-check the re-location of events prior to the inversion with the SIMUL2000 code for the synthetic tests, and indeed the events are relocated (as described in the manual)

with performing the step of initial source locations.

Another problem of synthetic modeling in the SIMULP workflow is that the anomalies are predefined in the same nodes as used for inversion. Successful restoring the anomalies centered in the nodes with spacing of 30x10 km gives an impression that the existing observation scheme would allow us to resolve such size of anomalies in the case of experimental data. Obviously, it might be true only if the anomalies perfectly centered with nodes. However, if an anomaly of 30x10 km size is located between nodes, it would obviously not be recovered, or strongly smeared. As the locations of real anomalies in the nature are not known, such modeling gives wrong assessment for the resolution. The shapes of synthetic models should be completely independent of the parameterization grid. In this case, we will recover not only amplitudes of anomalies (as in the present case), but also their locations and shapes, that is much more complicated.

As discussed above the inversion grid spacing is already above the limit of resolution. The synthetic models shown in Figure 7 (previous Figure 8) do not follow a simple geometry and are not fully aligned with the inversion grid. Figure 7, right panel, anomaly at 5 km depth shows a synthetic restoration test with two obliquely to the grid-orientated anomalies. One anomaly is at shallow depth (5km), the other one directly below (following the downgoing prolongation of the IFZ) at the depth of the plate interface. Of course one could make arbitrary anomalies with complex shapes, but since the anomalies are not known this would make the synthetic modelling overcomplicated. Furthermore if we would "smear" out an existing input anomaly across different grid nodes (e.g. not only using constant velocity perturbations of +/-5%) the restoration test would make a judgment of stability and spatial resolution more difficult.

Other specific comment on the paper. I did not understand the meaning of the 2D modeling performed in this study. Was it based on all data in the area? Does it mean that velocity along Y-coordinate is presumed to be constant? If yes, I would hardly expect any stable solution because of the existence of significant heterogeneities along

the trench line (for example, due to the presence of IFZ).

Why the Vp/Vs ratio is only shown for the 2D model, and not for the 3D model? It would be interesting to see the variations of the Vp/Vs ratio in the map view. I expect that if the IFZ is saturated by water, it would be seen in the Vp/Vs distributions.

[Figure]

As already stated the S arrivals are characterized by a higher noise level due to the tropical conditions and anthropogenic noise sources (e.g. large trees for the stations in the tropical forest, traffic close to settlements).

We added to chapter 3 (Local earthquake tomography):
*"We inverted 3D velocity models for vp/vs ratios. However, due to the limited data variance reduction of the 3D inversion of vp/vs, and the low quality of S onsets, the 3D vp/vs inversion is not robust. Therefore we only discuss vp/vs ratios of the 2D inversion."*

And modified the sentence in chapter 2 (earthquake data):

*"The main sources of noise on the records were tree movement, rain due to the tropical environment and anthropogenic noise (e.g. traffic), affecting in particular the horizontal components."*

We added to the first sentence of section 5.4 (vp/vs model of the forearc):
*"As discussed (chapter 3) S-onsets are of lower quality due to tropical conditions and anthropogenic noise. Therefore, we only present the 2D vp/vs inversion results (Figure 5, 8b, previous Figure 5 and 6)."*

In horizontal sections, it would be better to present relative anomalies instead of absolute velocity. In the present case, it is hard to see any nuances in dominantly green, yellow or red colors corresponding to absolute velocities at specific depths. The traditional "blue-white-red" scheme would provide much clearer images for the velocity variations.

On the use of the absolute velocity values:
Relative anomalies show the differences in respect to a given reference model. Since anomalies relative to a reference model are more difficult to understand we prefer absolute velocities, in agreement with other papers showing absolute velocities of local earthquake tomography. Obviously, figures using relative velocities enhance spatial variations in the velocity model. Due to the results of our checkerboard tests we think that these small variations are not resolved and prefer to show absolute velocities, which better reflect the resolution capabilities of our inversion.

On the choice of the colour scale:
We currently use the colour scheme rainbow which is a widely used colour scale for absolute vp velocities. This colour scale was previously used for various local earthquake tomography publications from different groups (e.g. DeShon et al., 2006, LET Nicaragua, doi: 10.1111/j.1365-246X.2005.02809.x; Pesicek et al. 2010, inversion of regional–global data, doi: 10.1111/j.1365-246X.2010.04630.x; Hicks et al., 2012, LET Central Chile, doi: 10.1016/j.epsl.2014.08.028). We optimized all superimposed

drawings (coastlines, Sumatran fault, events, Mentawai fault etc.) for optimal visibility. Therefore, we did not apply changes to the colour scales.

In vertical sections, in addition to absolute velocity, it would be helpful to present also the relative anomalies. In some cases, they appear to be very informative for interpretation.

See previous point.

P8L3: Does the 76% of reduction correspond to absolute residuals (L1) or squared values (L2)?

This corresponds to squared values.

Figure 6 is mentioned in the text prior Figure 5.

Previous Figure 5 is now Figure 8 and Figures 8,7 and 6 were renamed to Figures 7,6 and 5.

Was there any 3D synthetic modeling for the Vp/Vs ratio? Why the synthetic recovery results in the 3D case are shown only for the Vp, and not for the Vp/Vs?

We did extensive 3D synthetic modelling and checkerboard tests. As discussed above the resolution of the 3D vp/vs velocity model is poor and the residual reduction of the 3D vp/vs inversion is small. Therefore, we did not include the 3D vp/vs velocity model with their corresponding synthetic models and checkerboard tests. The 2D vp/vs checkerboard test is shown in Figure 5, right panels.

In Figure 8, the contours of the initial synthetic model should be highlighted.

The contours of the initial synthetic model in Figure 7 (previous Figure 8) are already highlighted in red and green colour using the same colour scale as the recovered models. The caption states already:
*"Red and green lines indicate the 5% contour lines of the input anomalies."*

---

## Author Response (AR1)

[revised manuscript text omitted]

---

## Author Response (AR2)

Thank you very much for your diligent consideration of the reviewers' concerns. It gives me great pleasure to state that the Topical Editor and I are happy for this paper to be published subject to a few very minor technical corrections.

Here are the few very minor (hence labeled technical) additions coming out of your reply to Koulakov's review that the Topical Editor and I would like you to incorporate for the final manuscript:

- Please mention in the text that source localisation is done, and where in the workflow this takes place. Readers should not need to consult the manual to understand what kind of basic operations the code is performing.

We added this statement to the third paragraph on page 7:
*"……pick uncertainties and all events were relocated prior to each iteration."*

- In the discussion, please also mention the limitations and subjective choices that are necessary to utilise this method, including for instance the limited scope of synthetic tests, choice of parameterisation (as in any tomographic approach).

Chapter 4 of the manuscript titled "Resolution and Uniqueness" is exclusively devoted to the limitations of the method. We added as first sentences of this chapter the following sentences (Page 9):

*"The method of LET tries to find a set of hypocentres and a velocity model, which jointly fit the arrival times best. Therefore, any LET code has some limitations, which include a finite number of synthetic recovery test and a partially subjective choice of parameterisation (e.g. grid spacing) of the velocity model or the choice of the damping value. As discussed in the previous chapter, SIMUL2000 uses a fixed velocity grid definition and a constant damping value set according to finding a compromise between obtaining a good data fit with low model variance, as judged by a trade-off curve.*

- "stable" 2D inversion: One shouldn't give readers the impression that a stable or robust, convergent inversion has more to do with reality than parameterisations or 3D for which inversions may become unstable: It may simply come from oversimplification of reality. Again, please make sure to mention something along these lines in the discussion, related to limitations of interpreting the 2D approximation rather than reality, and relate this to the uncertainty in interpreting tomographic results.

*We do not necessarily agree with the statement that a stable, robust and convergent inversion is only related to parameterisation and three-dimensional structural effects. In particular, for cases where the misfit becomes significantly smaller a robust inversion (here in the sense that there is little dependency on the 1D velocity input model) might possibly indicate that the model fits the data better (depending on the damping curve). In turn, we agree that oversimplification might result in more stable inversions. In the case of the Sumatra LET we did extensive testing of the inversion that we are convinced that the stability of the 2D inversion is most likely due to limited 3D heterogeneities since all 3D runs did not result in massively reduced data fits.*

*We modified this sentence on page 10:*

*"We then carefully checked the dependency of the 2D inversion on the velocity models and only found minor dependency of the 1D input model, indicating a very stable result of the 2D inversion, which suggests a well defined global minimum in the solution space for the 2D inversion. The independence of the inverted 2D velocity model on the 1D input models alone does not necessarily point to a better imaging capacity of the model and might be as well related to oversimplification of reality. We find this stability of the 2D inversion for different velocity model parameterisations (lateral and depth spacing) and a wide range of 1D vp velocity input models. Furthermore, the following 3D inversion only results in a modest further improvement of the fit. The trench-perpendicular velocity heterogeneity (2D structure) is thus more important than trench-parallel heterogeneity (3D structure)."*

it doesn't hurt to mention that you have done extensive 3D checkerboard tests and modeling even though the residual reduction is small.

We agree and modified the last paragraph of page 8:

*"We inverted 3D velocity models for vp/vs ratios and did extensive 3D vp/vs checkerboard tests, synthetic modelling and parameter test. However, due to the low quality of S onsets, the 3D vp/vs inversion was not robust and the data variance reduction was always small."*

Congratulations again, and many thanks for considering Solid Earth for your submission.

Best wishes
Huw

Thank you very much for the positive and constructive review process.

Best wishes
Dietrich Lange

[revised manuscript text omitted]